# Sports Elite Means Vaccine Elite? Concerns and Beliefs Related to COVID-19 Vaccines among Olympians and Elite Athletes

**DOI:** 10.3390/vaccines10101676

**Published:** 2022-10-08

**Authors:** Tomasz Sobierajski, Jarosław Krzywański, Tomasz Mikulski, Andrzej Pokrywka, Hubert Krysztofiak, Ernest Kuchar

**Affiliations:** 1Faculty of Applied Social Sciences and Resocialization, University of Warsaw, 26/28 Krakowskie Przedmieście Str., 00-927 Warsaw, Poland; 2National Centre for Sports Medicine, 63A Żwirki i Wigury Str., 02-091 Warszawa, Poland; 3Mossakowski Medical Research Institute, Polish Academy of Sciences, 5 Pawińskiego Str., 02-106 Warszawa, Poland; 4Department of Biochemistry and Pharmacogenomics, Medical University of Warsaw, 1 Banacha Str., 02-097 Warszawa, Poland; 5Department of Pediatrics with Clinical Assessment Unit, Medical University of Warsaw, 63A Żwirki i Wigury Str., 02-091 Warszawa, Poland

**Keywords:** professional athlete, immunization, pandemic, coach, social vaccinology

## Abstract

(1) Background: The purpose of this study was to investigate the concerns and beliefs of Olympians and elite athletes toward COVID-19 vaccination. (2) Methods: The study was framed by a quantitative method and was conducted using the PAPI (*pen and paper interview*) technique among 895 Polish elite athletes representing 34 sports. (3) Results: Three-quarters (76.3%) of the athletes were vaccinated against COVID-19; statistically participants were more likely to be women, and athletes who participated in the Olympic Games. Four in ten (39.2%) were in favor of vaccination. Athletes were mainly concerned that COVID-19 would exclude them from training/competition (19.3%) and could have a long-term impact on their health (17.2%). Athletes who were vaccinated reported much higher confidence in the composition of the vaccine and the doctors who recommended vaccination than unvaccinated athletes. Athletes who competed at the Olympic level were more likely than others to disbelieve that vaccines were produced too quickly and were not well tested. National-level athletes showed the highest degree of distrust in the government regarding COVID-19 vaccination, with one in six respondents distrusting doctors with respect to COVID-19 vaccination. Four in ten respondents said they were in favor of vaccination. (4) Conclusions: Athletes’ attitudes toward COVID-19 vaccination were significantly influenced by their environment—especially coaches and relatives. The power of social norms with respect to the decision to vaccinate against COVID-19 was very strong. Therefore, it is essential to build awareness about preventive policies among athletes and their social environment.

## 1. Introduction

The COVID-19 pandemic has had varying impacts on people from different walks of life, including sports in the sense of mass sports competitions [1]. Many major sporting events were canceled and others were postponed; those that ultimately took place, such as the Olympic Games in Tokyo, were held under a total sanitation regime and virtually without an audience [2,3,4]. The pandemic restrictions and the SARS-CoV-2 virus affected not only the participation of elite athletes in sporting events but also their daily training sessions, which either could not take place or occurred irregularly as a result of the closure of sports facilities and, above all, as a result of COVID-19 infections among athletes themselves, as well as their coaches and other colleagues [5]. In addition, any infection and deterioration of an athlete’s health in competitive sports could result in impeded athletic performance, which is why preventive healthcare in the form of vaccination is crucial for athletes [6,7].

The introdcution of COVID-19 vaccines represented hope for the world, including athletes, especially elite athletes, offering a chance to return to everyday life [8,9]. The launch of a global vaccination program resulted in the 2022 Beijing Winter Olympics and many other mass sports gatherings being able to take place as usual while adhering to the sanitary restrictions set by the organizers.

For most athletes who compete at the international level, the loss of one or two seasons due to a pandemic represents an irreparable loss, as in many sports, the peak of form does not last long [10,11,12]. Therefore, the COVID-19 vaccine represented a chance for such athletes to return to training, competing, winning, or simply working. In addition, elite athletes in most countries enjoy a very high level of public recognition, serving as authorities in terms of sports, as well as in terms of health and preventive health [13,14,15,16]. For this reason, the conduct of elite athletes may have had a direct or indirect impact on the pro- or anti-vaccination behavior of individuals and fans of athletes. In Poland, elite athletes were invited by the government campaign to encourage citizens to get vaccinated, with the hope that the power of their authority would increase the number of people who would get vaccinated. Dozens of Olympic athletes joined forces and called not only for COVID-19 vaccination but also for equal access to vaccination for all people in the world [17].

For this reason, we wanted to determine the attitudes of Olympic and elite athletes toward vaccination against COVID-19. We focused on several main research questions. We were interested in determining whether athletes want to be vaccinated, who they trust when it comes to vaccination, their attitudes toward the pandemic and COVID-19 vaccination, who influenced their vaccination decision, and how they perceive COVID-19 as a threat to their health and the development of their sports career.

## 2. Material and Methods

### 2.1. Study Design and Sample Size

The study was cross-sectional in nature, framed by a quantitative method, and conducted using the PAPI (*pen and paper interview*) technique.

The study included all National Team athletes subjected to routine periodic check-ups at the National Center for Sports Medicine (NCSM) in Warsaw between September and November 2021. CSM is a medical facility employing doctors, nurses, physiotherapists, psychologists, nutritionists, and biomedical specialists who care for the top Polish athletes on the National Team. Its task is to accompany athletes during their preparations for significant international sports competitions, including the Olympic Games.

The National Team consists of the top Polish athletes, who represent the country in international sports competitions and are affiliated, by discipline, with Polish sports associations. The total number of active athletes on the Polish National Team is 7536, and 1073 athletes took part in this research. For further analysis, 895 fully completed questionnaires were included. Therefore, the study sample is representative of athletes from the Polish National Team. Participation in the study was voluntary. The persons who distributed and collected the questionnaires from the athletes were trained for the implementation of this study and were knowledgeable about the purpose of the study, the distribution method of the research tool, and the principles of supervision with respect to data collection to ensure the anonymity of the respondents and confidentiality of the collected data.

### 2.2. The Questionnaire

The questionnaire used in the study was explicitly created for this research as part of the work of a multidisciplinary team of sociologists and medical doctors of various specialties, who are also authors of this publication. The questionnaire consisted of 28 questions, most of which were closed questions, with the remaining questions being semi-open for self-completion. Most of the questions were accompanied by list of answers from which one or more answers could be chosen according to the instructions placed next to the question. For some of the questions, a 5-point or 10-point scale was used. Additionally, some of the answers next to the questions had a filtering character and referred the respondents to subsequent questions depending on the nature of the answers.

The first part of the questionnaire consisted of demographic data, eliciting information about gender, age, height, weight, and discipline. The main part of the questionnaire consisted of questions about COVID-19 vaccination and associated attitudes and beliefs. The questionnaire also included questions about COVID-19 infection and its possible impact on training plans. The last part of the questionnaire was designed only for athletes who had been vaccinated against COVID-19. In this section, we asked, among other things, about physical complaints after receiving subsequent doses of the vaccine and the effect of receiving the vaccine on the implementation of training plans.

Before implementing the study, a pilot study was conducted among 30 randomly selected athletes from the Polish National Team. The pilot study was conducted to evaluate the tool and verify the categories, meanings, and methodological assumptions used in the tool. The pilot study participants did not make any comments on the tool. The questionnaire took the respondents an average of 20 min to complete.

### 2.3. Ethical Considerations

The ethics committee of the Medical University of Warsaw approved the study protocol (permission number AKBE/180/2021). Before conducting the study, participants were informed about the nature of the study and the confidentiality of the collected data. No personal data of the study participants were collected during the study. Instead, the collected data were transferred by a trusted researcher to a specially prepared confidential database prior to statistical analysis so that it was impossible to identify the individual respondents.

### 2.4. Statistical Analysis

Statistical analyses were carried out using IBM SPSS Statistics 27.0.1.0. Quantitative and categorical variables were described with the methods of descriptive statistics. The scales in the questionnaire were validated using Cronbach’s alpha test. Normality was calculated using the Shapiro–Wilk test. The chi-square test and Fisher’s exact test were used to compare independent categorical variables. Bivariate Pearson correlation coefficients were estimated to analyse construct validity. Considering the size of this sample and the number of athletes from the Polish National Team (*n* = 7536), the maximum error was 3% for a confidence level of 95% and a fraction size of 0.5. For analyses, a *p* values < 0.05 were considered statistically significant.

### 2.5. The Local Context of Vaccination

In Poland, COVID-19 vaccination for National Team athletes was not mandatory and was carried out according to the schedule provided by the National Vaccination Program, which was announced by the Polish government in December of 2019 [18]. The initial phase of the vaccination schedule consisted of four stages, according to which priority of vaccination was given to the elderly, sick people, and distinguished professionals, including healthcare workers, uniformed service members, and teachers. National Team athletes were not included in the vaccination schedule as a priority group to receive vaccinations earlier, so athletes who wanted to be vaccinated received COVID-19 vaccinations according to their competition scheduled from mid-2021 onwards at any time after registering for the vaccination. Only athletes who qualified for the Tokyo Olympics were allowed to be vaccinated earlier, in May 2021, with the JCOVDEN^®^ (Ad26.COV2.S) vaccine.

## 3. Results

### 3.1. Sample Demographic Characteristics

Most of the respondents were male (*n* = 509, 56.9%) and aged between 18 and 25 years (*n* = 490, 54.7%). The average age of the participants was 22 years. The largest group of respondents were those aged 18 (12.6%), followed by those aged 17 (10.1%) and those aged 19 (8.2%). Respondents represented 34 sports disciplines. The two most represented disciplines were athletics (*n* = 148, 17%) and handball (*n* = 143, 16%). Almost half of the respondents were international-level athletes (INT) who competed in World Championships and/or European Championships (*n* = 412, 46.1%), followed by national-level athletes (NAT) (*n* = 384, 43.1%) and Olympic athletes (OL) (including Paralympic Games athletes; *n* = 97, 10.8%) (Table 1).

### 3.2. Vaccination Status against COVID-19

Three out of four surveyed athletes were vaccinated (VA) against COVID-19 (76.3%, *n* = 683). One in four respondents had not been vaccinated against COVID-19 (NVA) (23.7%, *n* = 212). Among the VA group, not all were convinced to vaccinate from the beginning. In this group (*n* = 683), 34.8% (*n* = 238) of respondents were convinced “definitely from the beginning”, 34% (*n* = 232) of respondents were “rather” convinced, and 31.2% (*n* = 213) were “initially opposed but later changed their minds”.

Among the NVA (*n* = 212) group, one in five respondents (19.8%, *n* = 42) said they would get vaccinated shortly, two-thirds (66.5%, *n* = 141) said they did not intend to get vaccinated now but did not rule out vaccination in the future, and one in seven (13.7%, *n* = 29) said they would never get vaccinated.

Among men (*n* = 509), 72.1% (*n* = 367) were VA, and 27.9% (*n* = 142) were NVA; among women (*n* = 386), 81.9% (*n* = 316) and 18.1% (*n* = 70) were VA and NVA, respectively. Olympic-level athletes were the most vaccinated group, with 91.8% VA and 8.2% NVA in this group. Among international-level athletes, 76.8% were VA, and 23.2% were NVA; for national level-athletes, 71.9% and 28.1% were VA and NVA, respectively.

In terms of sports disciplines, among team sports athletes, soccer players were the least vaccinated against COVID-19 (six out of ten players), and handball and volleyball players were the most vaccinated group (three out of four players); among contact sports athletes, the lowest vaccination rate was reported among boxers (three out of ten athletes) (*p* < 0.001) (Table 2).

Those who declared that they had been vaccinated against COVID-19 did so mainly because they felt pressure from their coach (Figure 1). Moreover, those who had not been vaccinated against COVID-19 did not do so most often because their coach had advised them not to get vaccinated (Figure 2).

Seven in ten athletes surveyed (70.1%, *n* = 627) stated that their coaches were vaccinated against COVID-19. Only 3.7% (*n* = 33) stated that their coaches were not vaccinated, and one in four (26.3%, *n* = 235) did not know. Coaches of VA athletes were more likely to be vaccinated than those of NVA athletes (77.5% vs. 46.2%, respectively; *p* < 0.001) (Table 3).

One in four athletes (25.8%, *n* = 231) stated that all of their teammates were vaccinated against COVID-19. Four in ten (41.2%, *n* = 369) stated that most of their teammates were vaccinated, and one in five (20.8%, *n* = 186) stated that about half of their teammates were vaccinated. One in ten (9.9%, *n* = 89) stated that only some of their teammates were vaccinated. Among VA athletes, more teammates were likely to be vaccinated than among NVA athletes (33.2% vs. 1.9%, respectively; *p* < 0.001) (Table 4).

Four in ten respondents (43.6%, *n* = 390) stated that their relatives were vaccinated against COVID-19. One in four (26.9%, *n* = 241) stated that most of their relatives were vaccinated, and one in eight (12.3%, *n* = 110) stated that about half of their relatives were vaccinated. In one in ten cases (9.6%, *n* = 86), only some relatives were vaccinated, and in 6.1% (*n* = 55) of cases, no relatives were vaccinated against COVID-19. Significantly more relatives were not vaccinated at all among NVA respondents than among VA subjects (17.5% vs. 2.6%, respectively; *p* < 0.001) (Table 5).

Six in ten athletes (60.1%, *n* = 538) reported that their own opinion had the greatest influence on their decision to get vaccinated or not against COVID-19. The second largest influence on athletes in the context of getting vaccinated or not was relatives (17.5%, *n* = 157), and the third largest influence was other people (12.6%, *n* = 112). In 5.6% (*n* = 50) of cases, the decision was influenced by a coach, in 2.3% (*n* = 21) by a doctor, and in 1.9% (*n* = 17) by other athletes. NVA athletes were more likely to make this decision on their own than VA athletes (68.4% vs. 57.5%, respectively; *p* < 0.001) (Table 6).

### 3.3. General Attitude toward Vaccination

Four in ten respondents (39.2%, *n* = 351) considered themselves as rather in favor of vaccination, one in ten (10.0%, *n* = 89) declared being rather against vaccination, and one in two (50.8, *n* = 455) had no opinion on the subject. Among those who had not been vaccinated, only one in seven declared that he/she was rather in favor of vaccination (*p* < 0.001) (Table 7).

Four in ten respondents (42.1%, *n* = 376) stated that their coach was a supporter of vaccination. Only 2.3% (*n* = 21) stated that their coach was opposed to vaccination, and more than half (55.6%, *n* = 498) did not know. Among VA athletes, one in two said their coach was rather in favor of vaccination, whereas among NVA athletes, one in four coaches was reported to hold this view (*p* < 0.001) (Table 8).

### 3.4. Athletes’ Concerns and Beliefs Related to COVID-19 Vaccination and the COVID-19 Pandemic

Among the many concerns about COVID-19 vaccination, athletes most often reported being concerned that being vaccinated against COVID-19 may take them out of training or competition and may have long-term effects on their health (Table 9).

The intensity of the fears and beliefs of the athletes surveyed varies depending on whether the person was vaccinated or not (Table 10).

Athletes’ concerns and beliefs also varied in intensity according to the sports level at which the surveyed athletes competed (Table 11).

## 4. Discussion

There is no doubt that elite athletes, due to their activity involving high physical exertion and frequent close contact with other people, should receive COVID-19 vaccination, which will strengthen their immune system and help them fight COVID-19 if they get it [5]. Therefore, we examined the attitudes of Polish elite athletes, including Olympic team members, toward COVID-19 vaccination and the COVID-19 pandemic. We focused on whether their attitudes were associated with sport level and their vaccination status. Our study was exploratory. As the literature on the vaccination of elite athletes against COVID-19 is sparse, in the following discussion, we also suggest areas for further research that could be undertaken with respect to the vaccination behavior of elite athletes in a pandemic situation.

We found significant positive associations between vaccination and the vaccination of teammates, coaches, and relatives, which may be related to the effect of social norms. In a situation in which most people in the immediate environment are vaccinated, i.e., COVID-19 vaccination is the norm, athletes may feel pressure to get vaccinated [19]. Motivation to get vaccinated was therefore strongly linked to acceptance of norms held by the reference group, in this case, the team, squad, or family [20,21]. The aforementioned social activities related to vaccination are part of the sociological subdiscipline of social vaccinology, which describes people’s attitudes and actions toward vaccination [22]. Vaccination against COVID-19 is frequently socially contested due to the accepted intersubjective interpretation of treating COVID-19 as a benign disease. If an individual, in this case an athlete, finds himself in the field of influence of people who adhere to this interpretation of the social world, the chance of being vaccinated decreases [22].

The attitudes of vaccinated and unvaccinated athletes concerning fears and beliefs about the COVID-19 pandemic and COVID-19 vaccination, the relevance of which we demonstrated in this study, fit in with the theory of planned behavior (TPB), which is used to explain health behaviors, especially those over which people can exert self-control [23,24,25,26].

Regarding COVID-19 vaccination, it is important to note several constructs specific to TPB. One is behavioral intention, which focuses on what might motivate a given athlete to be immunized against COVID-19. Another is subjective norms, referring to the appropriateness of vaccination as a medical procedure and one of the elements of preventive healthcare. The next category is, as mentioned above, social norms. Power is also essential, which can be understood as the coach’s opinion regarding vaccination against COVID-19 [27,28]. Finally, another essential element is perceived behavioral control, which, in this case, is the availability (ease or difficulty) of immunization. The above categories determine the behavior of athletes, both those vaccinated and those who did not get vaccinate against COVID-19.

We demonstrated the impact of attitudes toward vaccination on getting immunized; supporters of vaccination, in general, were more likely to get immunized. Many people, both vaccinated and unvaccinated, do not have an educated attitude toward immunization, which may account for the deficiency in adult vaccination education in Poland, including among elite athletes. Vaccination for athletes should be essential to building their immunity as part of preventive health care [29].

Like the rest of the population, the surveyed athletes showed some concern about COVID-19 vaccination. As demonstrated by the study results, athletes tolerated immunization well, and their resting physiological indices return to pre-vaccination levels within four days of vaccination [30].

With respect to sports level, we showed a significant relationship concerning belief in the pandemic. One in ten athletes at the OL and NAT levels and one in six at the INT level do not believe in a pandemic, although at all levels, nearly half of the athletes do not doubt that a pandemic is occurring. The attitudes of athletes who doubt the existence of a pandemic are probably related to conspiracy theories and the fact that the athletes surveyed, as young people, use social media and the Internet as their main channels of information. These channels spread conspiracy theories with clear narrative structures and persuasive rhetoric [31,32,33].

In our study, we also demonstrated a significant relationship between sports levels and fear that a severe course of COVID-19 could exclude athletes from training (higher levels of fear were observed among NAT athletes), which is likely because elite athletes are at high risk of infection for several reasons. First, they travel frequently, sometimes to areas that do not have high vaccination rates in the community. Secondly, sports, even those not played as part of teams, tend to bring athletes into frequent contact with other athletes, coaches, and team members during training and competition. Thus, the risk of SARS-CoV-2 virus transmission among elite athletes, as with other droplet-transmitted diseases, is high [34]. In addition, intense exercise contributes to a reduction in the acquired immune response, known as the “open window”, which is responsible for increased risk of infections, exclusively respiratory diseases. In addition, based on the “open window” theory, 3 to 72 h after intense exercise, an infectious agent may be able to invade the host body, thus increasing the risk of opportunistic infections [29,35].

A significant relationship was also demonstrated between concerns of participant athletes about the long-term health effects of getting sick. In a retrospective study by Brawner et al., maximum exercise capacity was shown to be independently and inversely related to the likelihood of having COVID-19 [36]. Further research should be conducted to determine whether a lower risk of complications from COVID-19 overtreatment is associated with improved maximum exercise capacity [36]. Furthermore, endurance athletes may be susceptible to several complications of COVID-19, such as myocarditis, lung damage, coagulopathy, or pulmonary embolism, which may prevent an elite athlete from returning to competitive sports [37]. For example, a meta-analysis by Modica et al. found that the incidence of myocarditis among the athlete population ranges from 1% to 4% [38]. In a retrospective cohort study of a small group of elite athletes, including National Team and Olympic athletes who experienced mild or asymptompatic COVID-19 infection, there were no signs of acute myocarditis, but 19% of athletes had abnormalities assessed by cardiac MR [39].

### Limitations

The study presented here is subject to some limitations that are worth mentioning. The percentage of athletes who have been immunized against COVID-19 is higher than the percentage of people who have been vaccinated in the overall Polish population, as well as the percentage of people who have been vaccinated in Europe [40] because many athletes were “forced” to do so; otherwise, they would not have been able to participate internationally. The vaccination may have been lower if major sporting events had not been hosted during the pandemic (summer and winter Olympic Games, World and Continental Championships). For this reason, athletes’ attitudes toward vaccination may have been internalized.

An additional problem at the level of discussion is the small (even negligible) number of publications describing studies of elite athletes’ attitudes toward COVID-19 vaccination.

## 5. Conclusions

In the study, we showed that athletes’ attitudes toward COVID-19 vaccination are heavily influenced by the environment in which they live—especially coaches and relatives. The power of social norms with respect to the decision to get vaccinated against COVID-19 was obvious. For this reason, it is essential to build awareness about preventive healthcare, of which vaccination is a crucial component, not only among athletes but also in their social environment.

Our survey also indicated a strong need for health education among athletes with respect to preventing infectious diseases that can exclude them from training. Education and the formation of pro-health attitudes can increase the percentage of vaccinated elite athletes.

## Figures and Tables

**Figure 1 vaccines-10-01676-f001:**
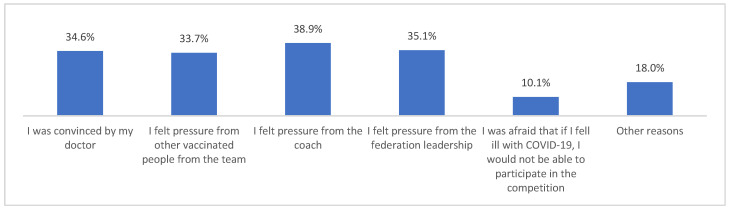
For what reason did you get vaccinated against COVID-19? (*n* = 683). Multiple-choice responses were presented in the questionnaire only for those who declared that they had been vaccinated against COVID-19.

**Figure 2 vaccines-10-01676-f002:**
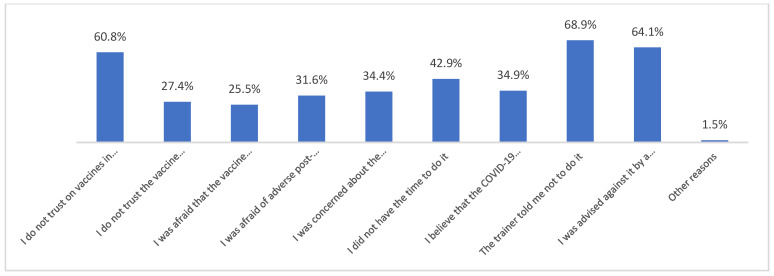
For what reason did you not get vaccinated against COVID-19? (*n* = 212) Multiple-choice responses were presented in the questionnaire only for those who declared that they had not been vaccinated against COVID-19.

**Table 1 vaccines-10-01676-t001:** Demographic characteristics of respondents (*n* = 895).

Variable	Female		Male		Total	
	N	%	N	%	N	%
Gender						
	386	43.1	509	56.9	895	100
Age (years)						
Up to 17	89	10.0	111	12.4	200	22.4
18–25	218	24.3	272	30.4	490	54.7
26–34	65	7.3	96	10.7	161	18.0
35 and over	14	1.5	30	3.4	44	4.9
Discipline						
Athletics	80	8.9	68	7.6	148	16.5
Handball	56	6.3	87	9.7	143	16.0
Skiing	12	1.3	40	4.5	52	5.8
Volleyball	10	1.1	37	4.1	47	5.3
Judo	23	2.6	21	2.3	44	4.9
Canoeing	8	0.9	32	3.6	40	4.5
Weightlifting	9	1.0	29	3.2	38	4.2
Wrestling	13	1.5	24	2.7	37	4.1
Rowing	16	1.8	14	1.6	30	3.4
Modern pentathlon	15	1.7	13	1.5	28	3.1
Soccer	0	0	27	3.0	27	3.0
Biathlon	12	1.3	13	1.5	25	2.8
Boxing	14	1.6	10	1.1	24	2.7
Basketball	24	2.7	0	0	24	2.7
Swimming	11	1.2	12	1.3	23	2.6
Fencing	15	1.7	8	0.9	23	2.6
Cycling	10	1.1	12	1.3	22	2.5
Sailing	8	0.9	13	1.5	21	2.3
Speed skating	9	1.0	11	1.2	20	2.2
Archery	5	0.6	8	0.9	13	1.5
Badminton	4	0.4	5	0.6	9	1.0
Curling	4	0.4	5	0.6	9	1.0
Snowboarding	4	0.4	2	0.2	6	0.7
Shooting	2	0.2	4	0.4	6	0.7
Sport climbing	4	0.4	2	0.2	6	0.7
Bobsleigh	4	0.4	2	0.2	6	0.7
Wushu	4	0.4	1	0.1	5	0.6
Taekwondo	1	0.1	3	0.3	4	0.4
Table tennis	2	0.2	2	0.2	4	0.4
Skateboarding	1	0.1	2	0.2	3	0.3
Artistic gymnastics	3	0.3	0	0	3	0.3
Figure skating	2	0.2	1	0.1	3	0.3
Skeleton	0	0	1	0.1	1	0.1
Tennis	1	0.1	0	0	1	0.1
Sports status						
Olympic level	48	5.3	49	5.5	10.8	10.8
International level	179	20.0	235	26.3	46.3	46.3
National level	159	17.8	225	25.1	42.9	42.9

**Table 2 vaccines-10-01676-t002:** Vaccinated (VA, *n* = 683) and unvaccinated (NVA, *n* = 212) against COVID-19 by sports discipline.

Sports Discipline			*p*-Value
	N (%)	N (%)	
Athletics	126 (85.1)	22 (14.9)	<0.001
Handball	112 (78.3)	31 (21.7)	
Skiing	49 (90.7)	3 (9.3)	
Volleyball	37 (78.7)	10 (21.3)	
Judo	35 (79.5)	9 (20.5)	
Canoeing	18 (45.0)	22 (55.0)	
Weightlifting	10 (26.3)	28 (73.7)	
Wrestling	23 (62.2)	14 (37.8)	
Rowing	26 (86.7)	4 (13.3)	
Modern pentathlon	20 (71.4)	8 (28.6)	
Soccer	16 (59.3)	11 (40.7)	
Biathlon	25 (100.0)	0 (0.0)	
Boxing	7 (29.2)	17 (70.8)	
Basketball	17 (70.8)	7 (29.2)	
Swimming	18 (78.3)	5 (21.7)	
Fencing	18 (78.3)	5 (21.7)	
Cycling	21 (95.5)	1 (4.5)	
Sailing	16 (76.2)	5 (23.8)	
Speed skating	18 (90.0)	2 (10.0)	
Archery	13 (100.0)	0 (0.0)	
Badminton	9 (100.0)	0 (0.0)	
Curling	7 (77.8)	2 (22.2)	
Snowboarding	6 (100.0)	0 (0.0)	
Shooting	6 (100.0)	0 (0.0)	
Sport climbing	6 (100.0)	0 (0.0)	
Bobsleigh	6 (100.0)	0 (0.0)	
Wushu	3 (60.0)	2 (40.0)	
Taekwondo	2 (50.0)	2 (50.0)	
Table tennis	3 (75.0)	1 (25.0)	
Skateboarding	2 (66.7)	1 (33.3)	
Artistic gymnastics	3 (100.0)	0 (0.0)	
Figure skating	3 (100.0)	0 (0.0)	
Skeleton	1 (100.0)	0 (0.0)	
Tennis	1 (100.0)	0 (0.0)	

**Table 3 vaccines-10-01676-t003:** Is your coach vaccinated against COVID-19? Responses broken down by vaccinated (VA, *n* = 683) and unvaccinated (NVA, *n* = 212) against COVID-19.

	VA	NVA	*p*-Value
	*n* (%)	*n* (%)	
Yes	529 (77.5)	98 (46.2)	<0.001
No	15 (2.2)	18 (8.5)	
I do not know	139 (20.4)	96 (45.3)	

**Table 4 vaccines-10-01676-t004:** Are your teammates vaccinated against COVID-19? Responses broken down by vaccinated (VA, *n* = 683) and unvaccinated (NVA, *n* = 212) against COVID-19.

	VA	NVA	*p*-Value
	*n* (%)	*n* (%)	
Yes, everyone	227 (33.2)	4 (1.9)	<0.001
Yes, most of them	314 (46.0)	55 (25.9)	
About half and half	101 (14.8)	85 (40.2)	
No, only some of them	34 (5.0)	55 (25.9)	
No, no one	0 (0.0)	9 (4.2)	
I do not know	7 (1.0)	4 (1.9)	

**Table 5 vaccines-10-01676-t005:** Are your relatives (family and friends) vaccinated against COVID-19? Responses broken down by vaccinated (VA, *n* = 683) and unvaccinated (NVA, *n* = 212) against COVID-19.

	VA	NVA	*p*-Value
	*n* (%)	*n* (%)	
Yes, everyone	373 (54.6)	17 (8.0)	<0.001
Yes, most of them	191 (28.0)	50 (23.6)	
About half and half	60 (8.8)	50 (23.6)	
No, only some of them	34 (5.0)	52 (24.5)	
No, no one	18 (2.6)	37 (17.5)	
I do not know	7 (1.0)	5 (2.8)	

**Table 6 vaccines-10-01676-t006:** Who had the greatest influence on your decision to get vaccinated or not against COVID-19? Responses broken down by vaccinated (VA, *n* = 683) and unvaccinated (NVA, *n* = 212) against COVID-19.

	VA	NVA	*p*-Value
	*n* (%)	*n* (%)	
Me	393 (57.5)	145 (68.4)	<0.001
Coach	46 (6.7)	4 (1.9)	
Doctor	19 (2.8)	2 (0.9)	
Other athletes	16 (2.3)	1 (0.5)	
Family/friends	130 (19.0)	27 (12.7)	
Others	79 (11.7)	33 (15.6)	

**Table 7 vaccines-10-01676-t007:** Athletes’ attitudes toward COVID-19 vaccination broken down by vaccinated (VA, *n* = 683) and unvaccinated (NVA, *n* = 212) against COVID-19.

	VA	NVA	*p*-Value
	*n* (%)	*n* (%)	
I’m rather a supporter of vaccination	322 (47.1)	29 (13.7)	<0.001
I’m rather opposed to vaccination	37 (5.4)	52 (24.5)	
I have no opinion	324 (47.5)	131 (61.8)	

**Table 8 vaccines-10-01676-t008:** Coaches’ attitudes toward COVID-19 vaccination among athletes vaccinated (VA, *n* = 683) and unvaccinated (NVA, *n* = 212) against COVID-19.

	VA	NVA	*p*-Value
	*n* (%)	*n* (%)	
He/she is rather a supporter of vaccination	321 (47.0)	55 (25.9)	<0.001
He/she is rather opposed to vaccination	14 (2.0)	7 (3.3)	
I have no idea	348 (51.0)	150 (70.8)	

**Table 9 vaccines-10-01676-t009:** Athletes’ concerns and beliefs related to COVID-19 vaccination and COVID-19 pandemic broken town in terms of frequency (*n* = 895).

Mean	SE	SD	Percentile			TD—Totally Disagree, RD—Rather Agree, NA/D—Neither Agree Nor Disagree, RA—Rather Agree, TA—Totally Agree
			25th	50th	75th	
I do not trust the government on COVID-19 vaccination.						
3.10	0.040	1.208	2.00	3.00	4.00	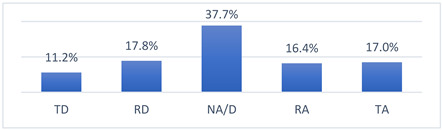
I do not trust doctors on COVID-19 vaccination.						
2.51	0.039	1.153	2.00	3.00	3.00	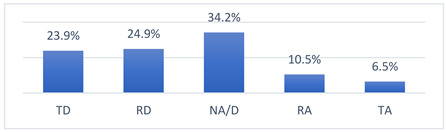
I do not believe in the effectiveness of the COVID-19 vaccine.						
2.52	0.041	1.212	1.00	3.00	3.00	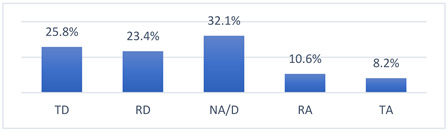
I am concerned about the long-term side effects of the COVID-19 vaccine.						
2.81	0.045	1.338	2.00	3.00	4.00	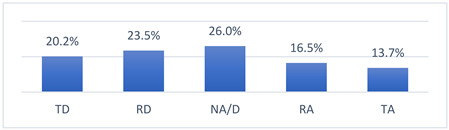
I do not trust the composition of the COVID-19 vaccine.						
2.80	0.042	1.253	2.00	3.00	4.00	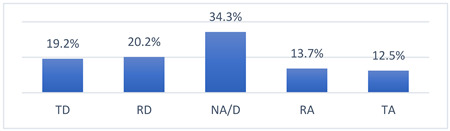
I believe that the COVID-19 vaccines were produced too quickly.						
2.96	0.043	1.293	2.00	3.00	4.00	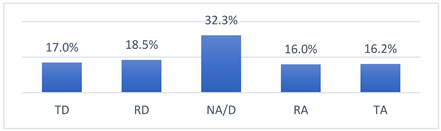
I believe that the COVID-19 vaccines have not been tested well enough.						
2.97	0.042	1.262	2.00	3.00	4.00	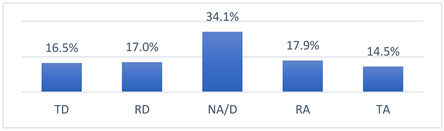
I believe that COVID-19 is not a dangerous disease.						
2.48	0.040	1.206	1.00	2.00	3.00	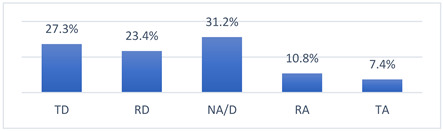
I do not believe in a COVID-19 pandemic.						
2.10	0.040	1.221	1.00	2.00	3.00	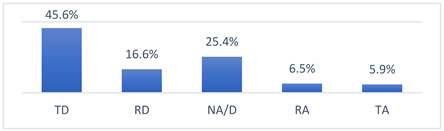
Mandatory vaccinations limit civil liberties.						
3.07	0.045	1.333	2.00	3.00	4.00	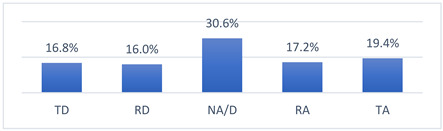
I am concerned about a severe COVID-19 infection that will exclude me from training/working/competition.						
3.05	0.045	1.348	2.00	3.00	4.00	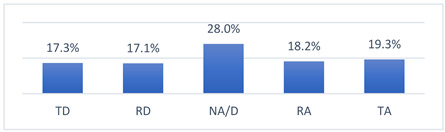
Getting over COVID-19 could have long-term effects on my health.						
3.24	0.040	1.196	3.00	3.00	4.00	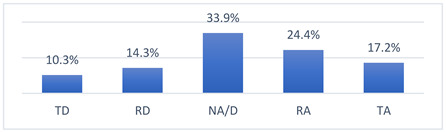

**Table 10 vaccines-10-01676-t010:** Athletes’ concerns and beliefs related to COVID-19 vaccination and the COVID-19 pandemic broken down by vaccinated and unvaccinated (*n* = 895).

	Strongly Disagree	Rather Disagree	Neither Disagree and Agree	Rather Agree	Strongly Agree	*p*-Value df x^2^	(95% CI) Mean	SE	SD	MD
	*n* (%)									
I do not trust the government on COVID-19 vaccination.										
VA	79 (11.6)	129 (18.9)	272 (39.8)	114 (16.7)	89 (13.0)	<0.001 4 32.699	(2.92–3.09) 3.01	0.044	1.158	3.00
NVA	21 (9.9)	30 (14.2)	65 (30.7)	33 (15.6)	63 (29.7)		(3.23–3.59) 3.41	0.094	1.312	3.00
I do not trust doctors on COVID-19 vaccination.										
VA	191 (28.0)	181 (26.5)	216 (31.6)	65 (9.5)	30 (4.4)	<0.001 4 50.343	(2.27–2.44) 2.36	0.043	1.116	2.00
NVA	23 (10.8)	42 (19.8)	90 (42.5)	29 (13.7)	28 (13.2)		(2.83–3.14) 2.99	0.078	1.142	3.00
I do not believe in the effectiveness of the COVID-19 vaccine.										
VA	209 (30.6)	177 (25.9)	207 (30.3)	59 (8.6)	31 (4.5)	<0.001 4 93.405	(2.22–2.39) 2.31	0.043	1.128	2.00
NVA	22 (10.4)	32 (15.1)	80 (37.7)	36 (17.0)	42 (19.8)		(3.04–3.37) 3.21	0.084	1.222	3.00
I am concerned about the long-term side effects of the COVID-19 vaccine.										
VA	163 (23.9)	178 (26.1)	184 (26.9)	98 (14.3)	59 (8.6)	<0.001 5 89.503	(2.49–2.69) 2.59	0.049	1.278	3.00
NVA	18 (8.5)	32 (15.1)	49 (23.1)	50 (23.6)	63 (29.7)		(3.33–3.68) 3.51	0.089	1.290	4.00
I do not trust the composition of the COVID-19 vaccine.										
VA	153 (22.4)	161 (23.6)	240 (35.1)	74 (10.8)	55 (8.1)	<0.001 4 95.392	(2.50–2.67) 2.59	0.045	1.180	3.00
NVA	19 (9.0)	20 (9.4)	67 (31.6)	49 (23.1)	57 (26.9)		(3.33–3.66) 3.50	0.085	1.233	3.50
I believe that the COVID-19 vaccines were produced too quickly.										
VA	132 (19.3)	144 (21.1)	242 (35.4)	94 (13.8)	71 (10.4)	<0.001 4 96.977	(2.66–2.84) 2.75	0.046	1.215	3.00
NVA	20 (9.4)	22 (10.4)	47 (22.2)	49 (23.1)	74 (34.9)		(3.46–3.81) 3.81	0.090	1.308	4.00
I believe that the COVID-19 vaccines have not been tested well enough.										
VA	132 (19.3)	135 (19.8)	245 (35.9)	114 (16.7)	57 (8.3)	<0.001 4 107.516	(2.66–2.84) 2.75	0.045	1.188	3.00
NVA	16 (7.5)	17 (8.0)	60 (28.3)	46 (21.7)	73 (34.4)		(3.51–3.84) 3.67	0.085	1.236	4.00
I believe that COVID-19 is not a dangerous disease.										
VA	209 (30.6)	165 (24.2)	200 (29.3)	70 (10.2)	39 (5.7)	<0.001 4 27.644	(2.27–2.45) 2.36	0.045	1.180	2.00
NVA	35 (16.5)	44 (20.8)	79 (37.3)	27 (12.7)	27 (12.7)		(2.68–3.01) 2.84	0.084	1.220	3.00
I do not believe in a COVID-19 pandemic.										
VA	342 (50.1)	117 (17.1)	157 (23.0)	39 (5.7)	28 (4.1)	<0.001 4 38.363	(1.88–2.05) 1.96	0.044	1.152	1.00
NVA	66 (31.1)	32 (15.1)	70 (33.0)	19 (9.0)	25 (11.8)		(2.37–2.73) 2.55	0.091	1.328	3.00
Requiring vaccinations limits civil liberties.										
VA	135 (19.8)	122 (17.9)	224 (32.8)	104 (15.2)	98 (14.3)	<0.001 4 71.477	(2.77–2.96) 2.87	0.050	1.296	3.00
NVA	15 (7.1)	21 (9.9)	50 (23.6)	50 (23.6)	76 (35.8)		(3.54–3.88) 3.71	0.086	1.246	4.00
I am concerned about a severe COVID-19 infection that will exclude me from training/working/competition.										
VA	109 (16.0)	111 (16.3)	190 (27.8)	136 (19.9)	137 (20.1)	0.046 4 9.696	(3.02–3.22) 3.12	0.051	1.338	3.00
NVA	46 (21.7)	42 (19.8)	61 (28.8)	27 (12.7)	36 (17.0)		(2.65–3.02) 2.83	0.094	1.362	3.00
Getting over COVID-19 could have long-term effects on my health.										
VA	56 (8.2)	86 (12.6)	230 (33.7)	183 (26.8)	128 (18.7)	<0.001 4 29.031	(3.27–3.44) 3.35	0.044	1.161	3.00
NVA	36 (17.0)	42 (19.8)	73 (34.4)	35 (16.5)	26 (12.3)		(2.71–3.04) 2.87	0.085	1.234	3.00

VA—vaccinated, NVA—nonvaccinated.

**Table 11 vaccines-10-01676-t011:** Athletes’ concerns and beliefs related to COVID-19 vaccination and COVID-19 pandemic by sports level (*n* = 895).

	Strongly Disagree	Rather Disagree	Neither Disagree and Agree	Rather Agree	Strongly Agree	*p*-Value df x^2^	(95% CI) Mean	SE	SD	MD
	*n* (%)									
I do not trust the government on COVID-19 vaccination.										
OL	11 (11.3)	17 (17.5)	41 (42.3)	15 (15.5)	13 (13.4)	0.296 8 9.583	(2.79–3.25) 3.02	0.117	1.155	3.00
INT	57 (13.8)	68 (16.4)	157 (37.9)	61 (14.7)	71 (17.1)		(2.93–3.17) 3.05	0.061	1.245	3.00
NAT	32 (8.3)	74 (19.3)	139 (36.2)	71 (18.5)	68 (17.7)		(3.06–3.30) 3.18	0.060	1.179	3.00
I do not trust doctors on COVID-19 vaccination.										
OL	24 (24.7)	23 (23.7)	34 (35.1)	10 (10.3)	6 (6.2)	0.301 8 9.516	(2.26–2.73) 2.49	0.117	1.156	3.00
INT	105 (25.4)	86 (20.8)	153 (37.0)	44 (10.6)	26 (6.3)		(2.40–2.63) 2.52	0.057	1.162	3.00
NAT	85 (22.1)	114 (29.7)	119 (31.0)	40 (10.4)	26 (6.8)		(2.39–2.61) 2.50	0.058	1.145	2.00
I do not believe in the effectiveness of the COVID-19 vaccine.										
OL	31 (32.0)	18 (18.6)	35 (36.1)	6 (6.2)	7 (7.2)	0.149 8 12.042	(2.14–2.62) 2.38	0.122	1.203	2.00
INT	109 (26.3)	86 (20.8)	137 (33.1)	43 (10.4)	39 (9.4)		(2.44–2.68) 2.56	0.061	1.245	3.00
NAT	91 (23.7)	105 (27.3)	115 (29.9)	46 (12.0)	27 (7.0)		(2.39–2.63) 2.51	0.060	1.179	2.00
I am concerned about the long-term side effects of the COVID-19 vaccine.										
OL	24 (24.7)	21 (21.6)	25 (25.8)	16 (16.5)	11 (11.3)	0.739 10 6.859	(2.41–2.95) 2.68	0.134	1.319	3.00
INT	78 (18.8)	102 (24.6)	115 (27.8)	60 (14.5)	59 (14.3)		(2.68–2.93) 2.81	0.064	1.297	3.00
NAT	79 (20.6)	87 (22.7)	93 (24.2)	72 (18.8)	52 (13.5)		(2.70–2.98) 2.84	0.071	1.387	3.00
I do not trust the composition of the COVID-19 vaccine.										
OL	24 (24.7)	16 (16.5)	33 (34.0)	11 (11.3)	13 (13.4)	0.795 8 4.639	(2.46–2.99) 2.72	0.134	1.321	3.00
INT	79 (19.1)	86 (20.8)	141 (34.1)	53 (12.8)	55 (13.3)		(2.68–2.93) 2.80	0.062	1.263	3.00
NAT	69 (18.0)	79 (20.6)	133 (34.6)	59 (15.4)	44 (11.5)		(2.69–2.94) 2.82	0.063	1.228	3.00
I believe that the COVID-19 vaccines were produced too quickly.										
OL	22 (22.7)	14 (14.4)	34 (35.1)	13 (13.4)	14 (14.4)	0.687 8 5.646	(2.56–3.09) 2.82	0.134	1.323	3.00
INT	68 (16.4)	74 (17.9)	138 (33.3)	64 (15.5)	70 (16.9)		(2.86–3.11) 2.99	0.064	1.292	3.00
NAT	62 (16.1)	78 (20.3)	117 (30.5)	66 (17.2)	61 (15.9)		(2.83–3.09) 2.96	0.066	1.288	3.00
I believe that the COVID-19 vaccines have not been tested well enough.										
OL	22 (22.7)	10 (10.3)	38 (39.2)	15 (15.5)	12 (12.4)	0.119 8 12.798	(2.59–3.10) 2.85	0.131	1.286	3.00
INT	67 (16.2)	73 (17.6)	149 (36.0)	63 (15.2)	62 (15.0)		(2.83–3.07) 2.95	0.062	1.256	3.00
NAT	59 (15.4)	69 (18.0)	118 (30.7)	82 (21.4)	56 (14.6)		(2.89–3.14) 3.02	0.064	1.263	3.00
I believe that COVID-19 is not a dangerous disease.										
OL	28 (28.9)	26 (26.8)	32 (33.0)	5 (5.2)	6 (6.2)	0.237 8 10.420	(2.10–2.56) 2.33	0.115	1.134	2.00
INT	111 (26.8)	96 (23.2)	140 (33.8)	41 (9.9)	26 (6.3)		(2.34–2.57) 2.46	0.057	1.167	2.50
NAT	105 (27.3)	87 (22.7)	107 (27.9)	51 (13.3)	34 (8.9)		(2.41–2.66) 2.54	0.064	1.264	2.50
I do not believe in the COVID-19 pandemic.										
OL	42 (43.3)	14 (14.4)	31 (32.0)	5 (5.2)	5 (5.2)	0.004 8 22.686	(1.90–2.38) 2.14	0.120	1.190	2.00
INT	176 (42.5)	56 (13.5)	119 (28.7)	36 (8.7)	27 (6.5)		(2.11–2.35) 2.23	0.062	1.263	2.00
NAT	190 (49.5)	79 (20.6)	77 (20.1)	17 (4.4)	21 (5.5)		(1.84–2.07) 1.98	0.059	1.168	2.00
Requiring vaccinations limits civil liberties.										
OL	20 (20.6)	13 (13.4)	33 (34.0)	17 (17.5)	14 (14.4)	0.068 8 14.566	(2.65–3.18) 2.92	0.133	1.312	3.00
INT	76 (18.4)	65 (15.7)	128 (30.9)	55 (13.3)	90 (21.7)		(2.91–3.18) 3.04	0.068	1.377	3.00
NAT	54 (14.1)	65 (16.9)	113 (29.4)	82 (21.4)	70 (18.2)		(3.00–3.26) 3.13	0.066	1.289	3.00
I am concerned about a severe COVID-19 infection that will exclude me from training/working/competition.										
OL	15 (15.5)	13 (13.4)	33 (34.0)	18 (18.6)	18 (18.6)	0.008 8 20.755	(2.85–3.38) 3.11	0.132	1.298	3.00
INT	79 (19.1)	62 (15.0)	135 (32.6)	71 (17.1)	67 (16.2)		(2.84–3.09) 2.96	0.065	1.317	3.00
NAT	61 (15.9)	78 (20.3)	83 (21.6)	74 (19.3)	88 (22.9)		(2.99–3.27) 3.13	0.071	1.391	3.00
Getting over COVID-19 could have long-term effects on my health.										
OL	6 (6.2)	8 (8.2)	37 (38.1)	31 (32.0)	15 (15.5)	0.040 8 16.148	(3.21–3.63) 3.42	0.107	1.049	3.00
INT	46 (11.1)	62 (15.0)	154 (37.2)	83 (20.0)	69 (16.7)		(3.05–3.28) 3.16	0.059	1.199	3.00
NAT	92 (10.3)	128 (14.3)	303 (33.9)	218 (24.4)	154 (17.2)		(3.15–3.40) 3.28	0.062	1.223	3.00

OL—Olympic-level athlete; INT—international-level athlete; NAT—national-level athlete.

## Data Availability

The data presented in this study are available upon request form the corresponding author.

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
