# Peer review of "Sports Elite Means Vaccine Elite? Concerns and Beliefs Related to COVID-19 Vaccines among Olympians and Elite Athletes"

_vaccines, 2022, doi:10.3390/vaccines10101676_

Round 1

Reviewer 1 Report

This is a reasonable paper with important data to add to the conversation on vaccination attitudes.  My most significant concerns are:

  1. The quantitative data are solid, but would be much enriched by qualitative discussions and analysis of focus groups or similar work with the athletes.  For example, it would be useful and relevant to know what the sources of the athletes’ information on the pandemic.  The authors state this is ‘probably social media’, but given the whole point—that they are elite athletes in an ‘elite athlete environment’-- it seems as or more likely they were receiving much information and education from their coaches, athletic associations, etc.  Related, it would be good to hear more about how the athletes’ responses compare to Poland more generally (there is only a hint of this), as well as analogous populations who have mandatory, or also non-mandatory, vaccination policies.

  2. English and other cultural references are awkward for an international audience.  The language is mostly understandable and easily editable, BUT in numerous instances results in things that are not clear scientifically or in other terms of results, which is a problem.  Some examples:

    1. The title: what does ‘Sports Elite Means Vaccine Elite?’ mean?  Unclear to me.

    2. Line 21: ‘4 in 10 were considered supporters of vaccines’.  ‘Considered’ is passive voice, suggesting the authors mean they, the authors, considered these athletes supporters of the vaccine, but that seems unlikely since the survey is asking whether the athletes themselves support the vaccine.

    3. Line 136: what are ‘yearbook vaccinations’?

    4. Line 217: ‘as rather supporters’ makes no sense

    5. Lines 257 and 299: what is a ‘sports class’?

  3. Many tables (1, 9, 10, and 11) and figures (1 and 2) are unnecessary or could be much clearer, significantly truncated and better visually shown, or moved to an appendix.

  4. The influence of coaches reads as contradictory, which should be restated or addressed more clearly in the introduction (e.g., line 26: athletes are ‘significantly influenced by their environment—especially coaches’ v. line 210 ‘In 5.6% (n=50) of cases the decision was influenced by a coach’).  This is the kind of nuance that could be complementarily addressed in focus groups with athletes.

  5. The Theory of Planned Behavior seems thrown in as an afterthought in a few sentences in the discussion.  Perhaps address this theory in the introduction and/or abstract?  What is this theory exactly and are there another theories that explain the data less well, etc?

Author Response

Thank you very much for your insightful analysis of our article and very interesting suggestions for further in-depth research and those that make our article more valuable.

Below are our responses to the various elements of the review.

This is a reasonable article with important data to add to the conversation about attitudes toward vaccination.  My most significant concerns are:

  1. the quantitative data is solid, but would be greatly enhanced by qualitative discussions and focus group analysis or similar work with athletes. For example, it would be useful and relevant to know the sources of athletes' information on pandemics. The authors state that these are "probably social media," but given the whole point - that these are elite athletes in an "elite athlete community" - it seems just as or more likely that they received a lot of information and education from their coaches, sports associations, etc.  That being said, it would be good to hear more about how the athletes' responses compare to Poland more generally (there's just a hint about that), as well as analogous populations that have mandatory or also optional vaccination policies.

Thank you very much for this suggestion. Because our study was exploratory and the first such a broad survey on vaccination, not only on COVID-19 vaccination among athletes from the Polish national team, we focused only on quantitative research. We wanted to explore what areas needed to be deepened with qualitative research.

However, the results from this study unquestionably lead us to the same conclusion: it is necessary to expand the knowledge on vaccination among athletes in the second round of research with qualitative research. The focus groups are an excellent indication, and we firmly believe that we will be able to implement this study. At the same time, we are glad that our research intuition also finds understanding in this tip.

  1. English and other cultural references are awkward for an international audience. The language is mostly understandable and easily editable, BUT in many cases it results in things that are not clear scientifically or in other performance categories, which is a problem. Some examples:
  2.  

  1. Title: what does "Sports elite means vaccine elite?" mean.Unclear to me.

Thank you for this indication. The title had two justifications: one local and the other global. The local context is that Polish elite athletes were Poland's primary ambassadors of COVID-19 vaccination. Their image was used in ads paid for by the government to encourage people to be vaccinated against COVID-19, an idea used in other countries for other vaccinations, as we write about in the Introduction. And this was the global context, as even the WHO reached out to elite athletes in their COVID-19 vaccination campaigns. It might seem that elite athletes' authority would translate into their health-promoting attitudes toward vaccination. And with this, as the results of our study show, it varies.

  1. Line 21: "4 in 10 were considered vaccine supporters."'Considered' is a passive voice, which suggests that the authors mean that they, the authors, considered these athletes to be supporters of vaccines, but this seems unlikely since the survey asks whether the athletes themselves support vaccines.

Thank you very much. This is an obvious error, and it has been changed. Lines: 21-22.

  1. Line 136: what are "vintage vaccinations"?

An error crept into the text. It referred to the prior inoculation of athletes in connection with their participation in competitions, such as the Olympics, which has been corrected. Thank you. Lines: 137-138.

  1. Line 217: "as rather in favor of" makes no sense

Has been changed to rather in favor of vaccination. Thank you. Line: 220.

  1. Lines 257 and 299: what is 'sport level'?

An error crept in; we meant sport level. This has been corrected.

  1. Many of the tables (1, 9, 10 and 11) and figures (1 and 2) are redundant or could be much clearer, much truncated and better shown visually or moved to the appendix.

We realize that the tables are extensive, and we had lengthy discussions about the appropriateness of using each table in the article. We excluded several other tables and charts from the article and left those that we believe are the most necessary and best describe the results. Such extensive tables result from adhering to the principle that what is in the tables is not described in the text, and vice versa.

5 The influence of coaches reads as contradictory, which should be repeated or addressed more clearly in the introduction (e.g., line 26: athletes are "significantly influenced by their environment - especially coaches" v. line 210 "In 5.6% (n=50) of cases, the decision was influenced by the coach").  This is the kind of nuance that could be complementarily addressed in focus groups with athletes.

Thank you for this indication. It has been completed. After the research, we also know that the role of coaches in the context of vaccination needs to be explored in more depth in qualitative research. Still, it is worth conducting quantitative and qualitative research among coaches on their attitudes toward vaccination.

6 The Theory of Planned Behavior seems to be thrown in as an afterthought in a few sentences in the discussion.  Perhaps this theory should be referred to in the introduction and/or summary?  What exactly is this theory and are there other theories that explain the data less well, etc.?

Thank you very much. The Theory of Planned Beahavior was not the primary theory we referred to in our analysis of the study; for this reason, we do not write about it more extensively in the methodology section. We relate some of the theory's assumptions to some attitudes. For this reason, there is only a small reference in the Discussion.

Once again, thank you very much for your insightful comments and observations.

Reviewer 2 Report

Congratulation to all authors for an interesting research project which is crucial for the policymakers and stakeholders to improve the uptake of COVID-19 vaccination.  However, the followings are the suggestions to improve the paper:

·       Abstract section background should have a clear sentence with an objective that is not very clear.

·       Study design is missing in both the abstract and methods section. Make this clear.

·       Abstract results sections only presented the descriptive findings which are missing the inferential findings (Table 11). State the key inferential findings as per the objectives in the abstract section.

·       Section 2.1 study design should clearly state which design and why this design.

·       How did you determine the sample size which is not clear at a moment?

·       How did you recruit study respondents?

·       In the result section, check and ensure Table 4 where some of the cells have 0 value which may affect the chi-square test value. 

·       Through reviewing the references and making references consistent which are mixed and matched in writing the first alphabet capital and small i.e. 1, 4…, missing place of publication i.e. 27.

Author Response

Thank you very much for your insightful analysis of our article and interesting suggestions that make our article more valuable.

Below are our responses to the various elements of the review.

I congratulate all the authors on an interesting research project that is of critical importance to policy makers and stakeholders to improve the prevalence of COVID-19 vaccination. However, the following are suggestions for improving the paper:

- The background of the Abstract section should have a clear sentence with a purpose that is not very clear.

Thank you very much. This has been corrected. Line 17.

- The study design is missing from both the abstract and the methods section. This should be clarified.

- In section 2.1, the study design should clearly state what design and why this one.

Thank you very much for these comments. This has been completed. Lines: 18-19 (Abstract), Lines: 82-83.

- In the results section of the abstract, only descriptive results are presented, with inferential results missing (Table 11). Please provide key inferential findings according to the objectives in the abstract section.

Thank you. This has been completed. Lines: 27-29

- How was the sample size determined, which is not clear at this time?

The sample size for the entire population of elite athletes in Poland with the specified maximum error and the adopted fraction size was 935, which was achieved after surveying 1073 athletes. 895 fully completed questionnaires were counted for further analysis.

- How were respondents recruited?

We have detailed information on this. In Lines: 79-85.

- In the results section, check and be sure to see Table 4, where some cells have a value of 0, which may affect the value of the chi-square test.

Thank you. We have checked this.

- By reviewing the references and making the references consistent, which are mixed and matched in writing the first uppercase and lowercase alphabet i.e. 1, 4..., missing place of publication i.e. 27.

Thank you very much. We have completed it.

Again, thank you very much for your insightful comments and observations.

Reviewer 3 Report

Interesting study on the influences for the decision of Polish athletes to get vaccinated or not. In total, 895 out of 7536 athletes responded to the questionnaire, with rather low numbers in several disciplines. How representative are the numbers for example in soccer, tennis or snow boarding? How do you explain the sex differences in the number of answering athletes?

Author Response

Thank you very much for your review and the questions posed. Below are our answers.

Interesting survey on the influences on the decision of Polish athletes to be vaccinated or not. A total of 895 out of 7536 athletes responded to the survey, with rather low numbers in several sports.

How representative are the numbers in soccer, tennis or snowboarding, for example?

The study was not a representative survey in the sense of a simple random draw from a pool of all athletes; for this reason and because some athletes were not available due to their preparation for competition, some sports are surveyed almost 100% - for example, national team soccer, handball, or volleyball, and others to a lesser extent.

The sample size for the entire population of elite athletes in Poland with the specified maximum error and the adopted fraction size was 935, which was achieved after surveying 1073 athletes. 895 fully completed questionnaires were counted for further analysis.

How to explain the gender differences in the number of responding athletes?

The higher number of men in the surveys we conducted is an almost perfect representation of the number of men and women in the national team in Poland. According to government estimates, about 40% are women, and 60% are men. At the same time, during the analysis, we found that gender does not have a statistically significant effect on attitudes toward COVID-19 vaccination.

Once again, thank you very much for your insightful comments and observations.

Round 2

Reviewer 1 Report

no further comments

Reviewer 3 Report

Clearly improved paper